# Bispecific Antibodies: A Smart Arsenal for Cancer Immunotherapies

**DOI:** 10.3390/vaccines9070724

**Published:** 2021-07-02

**Authors:** Gihoon You, Jonghwa Won, Yangsoon Lee, Dain Moon, Yunji Park, Sang Hoon Lee, Seung-Woo Lee

**Affiliations:** 1Department of Life Sciences, Pohang University of Science and Technology (POSTECH), Pohang 37673, Korea; ygh0824@postech.ac.kr (G.Y.); dain9730@postech.ac.kr (D.M.); 2ABL Bio Inc., Seongnam 13488, Korea; jonghwa.won@ablbio.com (J.W.); yangsoon.lee@ablbio.com (Y.L.); sang.lee@ablbio.com (S.H.L.); 3Biotechcenter, POSTECH, Pohang 37673, Korea; yunji@postech.ac.kr

**Keywords:** bispecific antibody, cancer immunotherapy

## Abstract

Following the clinical success of cancer immunotherapies such as immune checkpoint inhibitors blocking B7/CTLA-4 or PD-1/PD-L1 signaling and ongoing numerous combination therapies in the clinic,3 bispecific antibodies (BsAbs) are now emerging as a growing class of immunotherapies with the potential to improve clinical efficacy and safety further. Here, we describe four classes of BsAbs: (a) immune effector cell redirectors; (b) tumor-targeted immunomodulators; (c) dual immunomodulators; and (d) dual tumor-targeting BsAbs. This review describes each of these classes of BsAbs and presents examples of BsAbs in development. We reviewed the biological rationales and characteristics of BsAbs and summarized the current status and limitations of clinical development of BsAbs and strategies to overcome limitations. The field of BsAb-based cancer immunotherapy is growing, and more data from clinical trials are accumulating. Thus, BsAbs could be the next generation of new treatment options for cancer patients.

## 1. Introduction

Immunotherapy, including tumor-targeted monoclonal antibodies (mAbs) and immune checkpoint inhibitors (ICIs), has revolutionized anticancer therapy since the late 1990s. Currently, more than 30 different therapeutic mAbs have been developed and approved by the Food and Drug Administration (FDA) for oncology since the approval of rituximab, a B-cell-depleting anti-CD20 mAb used for the treatment of patients with B-cell non-Hodgkin lymphoma (NHL) [1,2]. Ipilimumab was the first FDA-approved ICI in 2011 for melanoma, which blocks cytotoxic T lymphocyte antigen-4 (CTLA-4), a negative regulator of T-cell immune function. Programmed cell death protein 1 (PD-1) is a checkpoint protein on immune cells and regulates the function of T cells by binding to programmed death-ligand 1 (PD-L1) on tumor cells. Anti-PD-(L)1 mAbs, which inhibit PD-1 or PD-L1, can block PD-1 signaling significantly and showed promising antitumor activity in treating certain cancers [3]. PD-1 inhibitors, nivolumab and pembrolizumab, were approved in 2014, and PD-L1 inhibitors, such as atezolizumab, avelumab, and durvalumab, were also approved in 2016 and 2017 for various cancers [4,5,6,7]. Currently, immunotherapies are widely used for the clinical treatment of various tumor types, including melanoma, non-small cell lung cancer, and colorectal cancer, and newly developed antibody-based therapeutics are under clinical investigation [8]. Although immunotherapies have shown significant and long-term efficacy in cancer patients, their clinical benefits in the overall population are still low because of limitations and challenges such as low response rate and resistance development [9,10]. In addition, immune-related adverse events (irAEs) caused by an excessively activated immune system, such as colitis, dermatological toxicity, hepatotoxicity, and pneumonitis, limit the clinical use of immunotherapies. Combination therapies seem to have improved efficacy compared to monotherapies but are also more likely to induce autoimmune-like toxicities, such as increased irAEs, which should be managed [11].

Advances in protein engineering technology have enabled the generation of various bispecific antibodies (BsAbs) that target multiple antigens as a single molecule [12]. BsAb-based immunotherapeutics may have the potential to improve clinical efficacy and safety. Therefore, the interest in the development of BsAbs has grown considerably, and there are various types of BsAbs in clinical and preclinical stages [13]. Since BsAbs can directly target two different antigens on immune cells or tumors (tumor-associated antigens; TAAs), BsAbs have an advantage over monospecific antibodies by synergistic inhibition of two different antigens or increasing cytotoxic activity. Currently, most BsAbs are T cell-redirecting antibodies, including two FDA-approved molecules (Blinatumomab and Catumaxomab) and an increasing number of next-generation bispecific or multi-specific antibodies. This review describes the classes, characteristics of BsAbs, biological rationales for developing BsAbs, current status and issues in the clinic, and strategies to overcome some limitations.

## 2. Binding Modules and Characteristics of Bispecific Antibodies

The characteristics of BsAbs, including physicochemical properties (size, stability, binding affinity, and valency), pharmacokinetics (PK)/pharmacodynamics (PD) properties, manufacturability, and immunogenicity, impact their efficacy, safety, and clinical success [14,15]. To achieve the desired biological characters, different BsAb formats with various modifications have been developed (Table 1).

### 2.1. Building Blocks of Bispecific Antibodies

A conventional immunoglobulin (Ig) G consists of two heavy chains (V_H_-C_H1_-C_H2_-C_H3_) and two light chains (V_L_-C_L_) connected by four disulfide bonds. Epitope binding domains consist of two pairs of variable domains of heavy and light chain, (V_H_ + V_L_)_2_, with mono-specificity. A single-chain variable fragment (scFv) consists of V_H_ and V_L_ connected by a short amino acid linker, while Fab refers to pairs of V_H_/C_H1_ and V_L_/C_L,_ including the constant domain [19]. Two scFvs with different binding specificity are tandemly linked by a short flexible linker (tandem scFv) in a bispecific T cell engager (BiTE) platform (Amgen, Thousand Oaks, CA, USA) [44]. Diabody refers to two non-covalently associated single chains of tandemly linked V_H_ and V_L_ with different specificity A and B (V_H_A-V_L_B and V_H_B-V_L_A). Immunoglobulin (IgG_2_ and IgG_3_ isotype) from camelids, also called heavy-chain only antibodies, has a unique structure devoid of the light chain component and the first constant domain of the heavy chain (C_H1_). These epitope binding domains of heavy-chain-only antibody are also called single variable domain, single-domain antibody (sdAb, V_H_H), or Nanobody [16,17] (Figure 1a).

### 2.2. Single-Chain Variable Fragment (scFv)-Based Bispecific Antibodies

The first scFv-based BsAb is a BiTE, in which two different scFvs are tandemly linked [44]. Blinatumomab, approved for the treatment of B-cell acute lymphoblastic leukemia (B-ALL), binds to CD3 and CD19 and elicits CD19-dependent T-cell activation [20]. Due to low molecular weight (~60 kDa), the half-life of BiTE is generally short, lasting less than several hours. To increase the in vivo half-life, the fragment crystallizable (Fc) domain was linked to BiTE. Other approaches to extend the half-life include ligating human serum albumin (HSA) itself or genetically linking the scFv against HSA [45,46] (Figure 1b).

### 2.3. IgG-Based Bispecific Antibodies

IgG-based BsAbs generally have a longer half-life compared with scFv- or domain-based antibodies comparable to conventional antibodies. A major challenge in IgG-based BsAbs is the proper pairing of two different heavy and light chains. It is possible to generate 16 different heterodimeric antibodies based on two different heavy chains and two different light chains without modification. The appropriate pairing of light chains with the corresponding heavy chains and the pairing of different heavy chains results in different heterodimeric antibodies. For heterodimerization of heavy chains, the knob-into-hole technology is generally used [32], but other technologies leveraging charge difference (e.g., Charge Repulsion Induces Bispecific; CRIB or substituting the C_H3_ region with T cell receptor (TCR) interface (used in Bispecific Engagement by Antibodies based on TCR; BEAT, Ichnos Biosciences, New York, NY, USA) are also in use [42,43]. For the pairing of heavy and light chains, diverse technologies have been developed, including (1) sharing the common light chains (Biclonics, Merus), (2) controlled Fab-arm exchange (DuoBody, Genmab), (3) switching the constant regions of heavy (C_H1_) and light chains (C_L1_) on one side of the heterodimer (CrossMAb, Roche), and (4) substituting the constant regions of immunoglobulins (C_H1_/C_L_) with the constant domains (C_α_/C_β_) of TCRs (WuXiBody, WuXi Biologics) [19,42]. Another approach is to maintain a regular IgG structure on one side but substitute the Fab with the scFv on the other side, as in the XmAb from Xencor such as vibecotamab (CD123 × CD3) and XmAb20717 (PD-1 × CTLA-4) [40,41] (Figure 1c).

Homodimeric bispecific antibodies are also frequently found in immunomodulating antibodies, especially in 4-1BB-engaging antibodies or dual immune checkpoint blocking antibodies. To induce TAA-dependent 4-1BB cross-linking and activation, 4-1BB-binding scFvs (ABL503, ABL111) [35,36] or anticalin (PRS-343) are connected to the C-terminal ends of the Fc of TAA-specific antibodies [37]. F-star Therapeutics develops its proprietary modular antibody technology platform, which utilizes some mutations in the Fc region, antigen-binding Fc fragments (Fcab). They are developing multiple pipelines of immuno-oncology therapies such as FS118 (PD-L1 × LAG-3) and FS222 (PD-L1 × 4-1BB) with its innovative BsAb technology, the tetravalent antibodies called mAb^2^ [47] (Figure 1c).

The Fc domain has effector functions mediated by their binding to the FcγR for IgG or to complement protein C1q [46], including antibody-dependent cellular cytotoxicity (ADCC), antibody-dependent cellular phagocytosis (ADCP), and complement-dependent cytotoxicity (CDC). Fc-engineering technology was introduced to generate versatile BsAb platform technology. Altering certain functions of the Fc domain of an antibody by substituting an amino acid sequence or glycoengineering can enhance or reduce effector functions [46,48]. Fc domains without Fc function or with retained/enhanced Fc function are generally used for immunomodulatory BsAbs and tumor-targeting BsAbs, respectively.

### 2.4. Affinity and Valency of Bispecific Antibodies

Depending on the immune cell target for agonistic activation, selecting the appropriate valency and affinity for each target is critical for TAA-dependent augmentation of the immune response and securing safety [15]. For CD3-targeting BsAbs, a single CD3-targeting arm is generally used to trigger TAA expression-dependent CD3 activation and T cell-mediated tumor killing while minimizing potential side effects from non-specific T-cell activation. Affinities of T-cell engagers to CD3 are diverse from 2 to 500 nM. Because CD3 is constitutively expressed in T cells, it is generally conceived that a lower binding affinity to CD3 might improve the PK, biodistribution, and safety [49,50]. To increase the TAA-binding activity yet minimizing the CD3 binding activity in the absence of TAA, two Fab regions targeting TAA and monovalent CD3 Fab were incorporated in a 2 + 1 format on the IgG backbone. Currently, both epcoritamab (CD20 × CD3, 1 + 1 format) and glofitamab (CD20 × CD3, 2 + 1 format) are in phase III clinical trials [31,33].

For 4-1BB-engagers, both one-arm and two-arm 4-1BB-engagers have been used to augment TAA-dependent 4-1BB trimerization and T cell activation. PRS-343 (HER2 × 4-1BB) [38], ABL503 (PD-L1 × 4-1BB) [35], and ABL111 (Claudin 18.2 × 4-1BB) [36] are bivalent 4-1BB engagers with a 2 + 2 format, while MCLA-145 (PD-L1 × 4-1BB) is a monovalent 4-1BB activator with a 1 + 1 format [28]. Tetravalent natural killer (NK) cell-engaging BsAbs, including AFM13, adopt the tandem diabody format, which contains a 9-amino acid linker that links variable domains of heavy and light chains to form a single polypeptide, and functionally active homodimers are generated by autonomous assembly of the corresponding V_H_ to V_L_ [25].

### 2.5. Immunogenicity of Bispecific Antibodies

The molecular structures of BsAbs are not naturally observed, raising the potential risks of immunogenicity. In addition to reducing drug availability and efficacy, the generation of anti-drug antibodies (ADAs) and neutralizing antibodies could also cause severe drug-related toxicities by forming drug/ADA immune complexes and eliciting hypersensitivity reactions [51]. Various factors affect the immunogenicity of antibodies, including product-related impurities, antibody origin (human, chimeric), dosing regimen, and target molecule [51]. Due to the short history of BsAb development in the clinics, reports on immunogenicity are limited: Less than 1% of patients treated with blinatumomab display ADA [51,52]. In the same way, no ADA-related issues were reported with other B cell-targeting BsAbs, such as mosunetuzumab (CD20 × CD3, 1 + 1 IgG format) and glofitamab (CD20 × CD3, 2 + 1 IgG format), regardless of their format [51]. This might be due to mechanism of action (MoA)-based phenomena in which the elimination of antibody-producing B cells by these BsAbs eliminates the possible generation of ADA. ADA generation was reported in other T cell-activating BsAbs such as PRS-343 (HER2 × 4-1BB using anticalin technology), where BsAbs at 2.5 mg/kg or more elicited ADA in 27.8% of cancer patients [53]. The ADAPTIR platform developed by Aptevo Therapeutics contains two different scFvs linked to the N- and C-terminal ends of the Fc domain. APVO-414 (PSMA × CD3) in the ADAPTIR platform elicited ADA in more than 50% of the patients, and further clinical development has been on hold [54,55]. AFM13 (CD30 × CD16A), a first-in-class tetravalent tandem diabody (TandAb)-based BsAb, showed ADA in 15 of the 28 patients treated, and half of the detected ADAs had neutralizing potential due to the chimeric origin of the anti-CD30 scFv [25]. It is not known whether tetravalent TandAb contributed to immunogenicity. Low ADA titers were observed in 42% of the patients treated with FS118. Interestingly, ADA by FS118 was transiently observed at higher dose levels and did not affect drug exposure [39]. V_H_H domains from camelids have high sequence identity with human type 3 VH domain and are expected to have low immunogenicity.

## 3. Multiple Types of Bispecific Therapeutics in Immuno-Oncology

Bispecific therapeutics can be divided into four major categories based on the biological types of targets and modes of action: (a) immune effector cell redirectors, (b) tumor-targeted immunomodulators, (c) dual immunomodulators, and (d) dual tumor-targeting BsAbs (Figure 2). Among these categories, both immune effector cell redirectors and tumor-targeted immunomodulators induce tumor-targeted activation of immune cells and are being developed extensively in the clinics. In both approaches, the selection of the optimal TAA is critical. The TAA expression pattern affects the safety and off-tumor toxicity because TAA is highly expressed in tumor cells or the tumor-associated environment, whereas it is absent in normal tissues. In addition, TAA expression will determine the potential indication of BsAbs due to tumor-specific expression. This review will focus on immune effector cell redirectors and tumor-targeted immunomodulators among BsAbs and briefly introduce the biological rationale for developing dual immunomodulators and dual tumor-targeting BsAbs in the last section.

## 4. Immune Effector Cell Redirectors

Among the four types of BsAbs, immune effector cell redirectors, especially T cell engagers, are the most advanced, with two approved products and several molecules in clinical development. These types of BsAbs include T cell-engagers, NK cell-engagers, and other immune cell-engagers composed of a TAA-targeting moiety and an immune cell-redirecting moiety.

### 4.1. T Cell-Engagers

T cells recognize target cells by detecting the cognate peptides presented by the major histocompatibility complex (MHC) through their TCRs under physiological conditions. The interaction between TCRs and the cognate peptide/MHC (pMHCs) brings the T cells and target cells in proximity and results in TCR clustering at the center of the immunological synapse. Following TCR clustering, cytolytic granules release cytolytic molecules, such as perforin and granzymes, into the target cells, eventually inducing target cell death. Cancer cells have several mechanisms to escape immunosurveillance, most of which are related to antigen processing and presentation in the context of MHC molecules.

T cell-engagers are BsAbs that engage TAAs and a component of the TCR complex (mostly CD3). They bypass TCR-pMHC interactions and facilitate the formation of cytolytic synapses between T cells and cancer cells, followed by polyclonal activation of T cells, thereby redirecting T cell cytotoxicity to the tumor cells. In this approach, T cell activation by BsAbs is independent of the TCR antigen specificity. Thus, it results in the activation of a large proportion of T cells even with low TAA expression and co-stimulation [56], although co-stimulation further enhances the activity [30]. T-cell-engaging BsAbs have been extensively explored in hematological and solid tumors for many years, and two such FDA-approved BsAbs, blinatumomab (Blincyto) and catumaxomab (Removab), have demonstrated favorable clinical outcomes. Although the latter was voluntarily withdrawn from the market for commercial reasons, there are currently more than 60 T-cell engagers in phase I/II clinical trials for the treatment of hematological and solid tumors.

Blinatumomab is a CD19- and CD3-targeting (CD19 × CD3) BsAb that the FDA approved in 2014 for the treatment of relapsed/refractory B-ALL [20], and is currently undergoing phase I/II clinical trials in other hematological malignancies such as diffuse large B-cell lymphoma and NHL [57]. The complete remission (CR) rate was 32%, although cytokine release syndrome (CRS) and neurotoxicity occurred in some patients [20]. Blinatumomab does not have an Fc region, thus having a short serum half-life of approximately 2 h [20]. Blinatumomab redirects T cell cytotoxicity at sub-picomolar concentrations by inducing immunological synapse formation between T cells and CD19^+^ target cells, activation and proliferation of T cells, and lysis of target cells [58]. Due to its short half-life, the dosing regimen for blinatumomab has been limited to continuous infusion in a clinical setting. For a prolonged half-life, some T cell engagers consist of the Fc domain, such as half-life extended BiTE and catumaxomab.

Catumaxomab is an EpCAM × CD3 BsAb, which was approved in 2009 for the treatment of malignant ascites in patients with ovarian cancer [59]. In contrast to blinatumomab, catumaxomab has a different molecular format. It consists of an asymmetric Fc-containing full-length Ab comprised of heavy and light chains binding to each target and produced from rat/mouse quadroma. The chimeric Fc domain binds to FcγRs on macrophages, dendritic cells (DCs), and NK cells and activates Fc-mediated functions to tumor cells [59]. However, since its Fc domain is not originated from human Ab, it could trigger immunogenicity and compromise its effectiveness in patients [60]. Other T-cell-redirecting BsAbs in clinical trials are shown in Table 2.

Liddy et al. [61] described a different type of BsAbs called ImmTACs (**I**mmune **m**obilizing **m**onoclonal **T**CR **A**gainst **C**ancer), consisting of a tumor-associated epitope-specific monoclonal TCR clone with picomolar affinity fused to an anti-CD3 scFv. ImmTAC uses an affinity-matured TCR that recognizes endogenously processed peptides bound to human leukocyte antigens (HLAs) on the cell surface. They generated several ImmTAC clones with the following epitope specificities: gp100_280–288_, MAGE-A3_168–176_, Melan-A/MART-1_26–35_, and NY-ESO-1_157–165_. These ImmTACs efficiently redirect T cells to kill cancer cells in vitro and in vivo in xenograft models [61]. An ImmTAC specific for the gp100 epitope, Tebentafusp (IMCgp100), is the most advanced molecule in the clinic to treat uveal melanoma patients. The overall survival rate at one year in phase I clinical trials was 73–74% [62]. ImmTAC molecules, such as IMC-C103C (anti-MAGE-A4 TCR × CD3), GSK01 (IMCnyeso, anti-NY-ESO-1 TCR × CD3), and IMC-F106C (anti-PRAME TCR × CD3), are under clinical investigation. This strategy suggests the applicability to intracellular antigens, which requires TCR optimization in the context of defined HLA haplotypes.

CD3 BsAb therapy is a form of immunotherapy that enables T cells to recognize and kill tumor cells. To prevent any potential bivalency-driven activation of T cells in the absence of TAAs, most formats for T cell engagers are preferably monovalent for CD3. In addition, T cell-engagers generally have a lower affinity for the CD3-targeting arm than the TAA-targeting arm. Although the binding affinity to CD3 is not thought to be correlated with T cell activation potential [50,63], a lower binding affinity to CD3 may improve the PK properties, biodistribution, and safety profile. Interestingly, CD3 BsAbs with different affinities targeting C-type lectin-like molecule 1 (CLL-1) have been tested: low (50 nM), high (0.5 nM), and very high (0.05 nM). In in vitro cell killing assays, the low-affinity CD3 BsAb showed lower potency than the variants with higher affinity. However, all variants led to a rapid depletion of CLL-1^+^ cells in transgenic mice, and low-affinity BsAb-treated mice showed a tendency toward a delayed rebound of target cells, suggesting a more sustained activity compared to higher affinity variants. PK analysis showed different clearances for the three variants, with the low-affinity BsAb showing the slowest clearance [64], and the affinity to CD3 also affects the biodistribution of BsAbs. In addition, HER2 × CD3 BsAbs with low-affinity (50 nM) or high-affinity (0.5 nM) CD3-targeting arms were evaluated in human CD3 transgenic mice. Treatment with HER2 × CD3 BsAb having a low-affinity arm led to a higher accumulation in HER2^+^ tumor tissues and a lower distribution to T-cell rich secondary lymphoid organs such as the spleen and lymph nodes [65].

Although T-cell engagers have shown superior efficacy in clinics, they still face hurdles such as low response rates in solid tumors and significant toxicity, including neurotoxicity and CRS, a systemic inflammatory response [66,67,68]. In a randomized trial, adverse events and serious adverse events, including neutropenia, infection, neurotoxicity, and CRS, were reported in ~99% and 62% of the patients, respectively [69]. Dose fractionation regimens have been suggested to mitigate CRS, which could be clinically manageable by administering steroids or tocilizumab (anti-IL-6R) [68]. In line with the previous affinity issue, the lower affinity of the anti-CD3 arm is related to decreased cytokine release potential. Mice receiving the BsAb variant with a high-affinity anti-CD3 arm showed elevated serum cytokines and proinflammatory mediators and developed vascular shock with fever. However, low-affinity BsAb-treated mice did not exhibit toxicity within 48 h and had variable increases in systemic cytokines that were generally much smaller than those seen in high-affinity BsAb-treated mice [64]. In another example, XmAb13551, which is a CD38 × CD3 BsAb (8 nM affinity for human CD3) against multiple myeloma (MM), showed effective MM cell clearance in mice [70] but triggered CRS at doses >0.2 mg/kg in monkeys [71]. Affinity-engineered XmAb13551 suggested that intermediate and very low-affinity variants were well tolerated at higher doses (0.5–3 mg/kg) and showed enhanced target cell depletion [71].

In addition to modifying the affinity of the anti-CD3 arm, the valency of the BsAb format may also affect its activity. Roche has developed IgG-like T-cell-engaging BsAbs with an asymmetric “2 + 1” structure with two TAA binders and a CD3 binder, and one TAA-binding Fab is linked to the N-terminus of the CD3-binding one. This format increased TAA avidity, enhanced tumor accumulation, and showed superior potency than the “1 + 1” version of the BsAb [72,73]. In clinical trials of a 2:1 CEA × CD3 BsAb (RG7802), 45% of patients showed disease control by the BsAb, and grade ≥ 3 related adverse events were observed in 28% of the patients [34], which was lower than that observed with blinatumomab [69]. The TAA-binding arm may hinder the CD3-binding arm, and TAA binding may unmask the CD3-binding arm. APVO-414 (PSMA × CD3) is an ADAPTIR BsAb format, which has bivalent arms for each target. APVO-414 showed ~30-fold higher cytotoxicity compared to the (scFv)_2_ format in vitro; however, cytokine production was much lower than that seen with the (scFv)_2_ format even when the same anti-CD3 Ab was used [74].

To achieve dual antigen-specific activation of CD3, a tandem (scFv)_3_ was split into two antigen-binding scFvs fused to either the V_H_ or the V_L_ domain of an anti-CD3 scFv, referred to as hemibody. When a complementary pair of hemibodies simultaneously binds to the respective antigens on a single target cell, the V_H_ and V_L_ domains reconstitute the original CD3-binding activity to engage T cells with dual antigen-positive tumor cells [75]. In addition, protease-activation of the anti-CD3 arm can be used for tumor-selective CD3 activation. The CD3-binding arm is masked by a N-terminally fused peptide [76] or anti-idiotypic anti-CD3 scFv [77] and unleashed in the tumor microenvironment (TME), which overexpresses several proteases, including matrix metalloproteinases. These strategies increase the maximum tolerated dose and lower the elevation of systemic cytokine levels compared to unmasked T-cell engagers.

### 4.2. NK Cell-Engagers

An alternative approach is activating and redirecting NK cells to tumor cells. NK-cell engagers may have a better safety profile than T-cell engagers while showing a similar clinical efficacy [78]. Several activating receptors can induce the cytotoxic functions of NK cells, including CD16 (FcγRIII), natural cytotoxicity receptors (NCRs; NKp30, NKp44, and NKp46), NKG2D (CD314), and DNAM-1 (CD226) [79,80]. CD16, a low-affinity FcγR with two isoforms, CD16A and CD16B, is the most common target for NK-cell engagement. CD16A (FcγRIIIA) is an activating receptor that is mainly expressed on NK cells and macrophages. CD16B (FcγRIIIB) is expressed on granulocytes and is not involved in killing tumor cells. When CD16A binds to the Fc domain of antibodies attached to its cognate antigen, it induces ADCC in antibody-binding cells without costimulation [81]. By activating CD16A, the BsAb can redirect CD16A-expressing effector cells to target cells, even without binding to the Fc region. Most NK-cell engagers have been developed for hematological malignancies.

AFM13 (CD30 × CD16A) is a tetravalent BsAb that targets CD16A and CD30, expressed in Hodgkin and Reed-Sternberg cells of patients with Hodgkin’s lymphoma [25]. AFM13 was well tolerated and showed an overall disease control rate of 61% (partial remission + stable disease) in patients with Hodgkin’s lymphoma, significant NK cell activation, and a decrease in soluble CD30 in peripheral blood during a phase I clinical trial [26] and is currently in a phase II trial. A CD16A-directed tri-specific, tetravalent antibody (BCMA × CD16A × CD200) termed aTriFlex showed dual antigen-selective NK cell-mediated lysis [82]. CD16-targeting BsAbs, which target CD33, CD133, or B7-H3, are further modified by introducing an IL-15 crosslinker, called TriKE, to induce expansion, priming, and survival of NK cells. It showed superior antitumor activity and enhanced survival of human NK cells in vitro compared to non-modified BsAbs [27,83,84].

Other examples of NK-cell engagers are NKp46- or NKG2D-targeting BsAbs, which have been investigated preclinically. NKp46 (NCR1, CD335) is expressed in NK cells, type 1 innate lymphoid cells, and a small population of T cells [79]. CD20 × NKp46 showed tumor regression in a xenograft model. In addition, by introducing an ADCC-enhanced Fc that can bind to CD16, tri-specific antibodies with more efficient ADCC activity than BsAbs were generated [85]. NKG2D is expressed on NK cells, CD8 T cells, and invariant natural killer T (iNKT) cells, and its activation relies on the expression of cognate ligands and induced-self proteins such as MICA and ULBP [80]. MM cell-targeting BsAbs were designed by targeting NKG2D and CS1 (CD319, SLAMF7). A CS1 × NKG2D BsAb induced cytotoxicity against CS1^+^ MM cells, IFN-γ production, and prolonged survival in a xenograft model of MM [86].

### 4.3. Other Immune Cell-Engagers

CD64 (FcγRI), a high-affinity FcγR, is expressed on hematopoietic cells such as monocytes, macrophages, activated neutrophils, and immature DCs [81]. CD64-engaging BsAbs were constructed via cross-linking of the anti-CD64 mAb (clone H22) Fab’ fragment with a TAA-targeting Fab’ fragment. The H22 clone reacts with an epitope distinct from the Fc region binding domain. However, no consistent antitumor activity [66,87] or toxicity [88] has been observed in clinical trials of CD64-engaging BsAbs. Additionally, iNKT cell-redirecting strategies using α-galactosylceramide-loaded CD1d extracellular domain fused with tumor-targeting scFv are also at an early preclinical stage [89].

## 5. Tumor-Targeted Immunomodulators

Similar to T cell-engagers, BsAbs targeting both TAAs and cosignaling molecules on T cells have also been extensively investigated. CD3 is expressed on all T cells, including resting T cells; however, costimulatory receptors, such as 4-1BB, are upregulated in tumor-specific tumor-infiltrating T cells, making them attractive targets for solid tumors [90]. Most cosignaling molecules are involved in either the immunoglobulin superfamily (IgSF) or tumor necrosis factor (TNF) receptor superfamily (TNFRSF) [91,92]. This section reviews tumor-targeted immunomodulatory BsAbs based on the superfamily to which their target belongs, and BsAbs in clinical trials are summarized in Table 3.

### 5.1. Immunoglobulin Superfamily

CD28 was the first identified costimulatory receptor on T cells and has been extensively characterized. Its ligands, B7-1 (CD80) and B7-2 (CD86), which also interact with CTLA-4 (CD152), are preferentially expressed on antigen-presenting cells (APCs). The B7/CD28 interaction probably remains the most potent costimuli for naïve T cells [91]. CD28 agonistic mAbs have been developed for therapeutic stimulation and induce potent T cell activation without TCR engagement [93]. However, one of these antibodies, TGN1412 (theralizumab), induced life-threatening CRS and multiple organ failure in clinical trials [94]. Several BsAbs have been developed for the controlled stimulation of CD28. r2820, a BsAb targeting CD28 and CD20, induces T cell activation in peripheral blood mononuclear cell (PBMC) cultures of healthy donors and CLL patients and kills CD20^+^ tumor cells. This proof-of-concept study showed the possibility of target cell-restricted activation of CD28 [95]. Two CD28-targeting BsAbs were developed, one for ovarian cancer (MUC16 × CD28) and another for prostate cancer (PSMA × CD28, REGN5678). Both BsAbs have limited activity but potentiate CD3 BsAbs to activate T cell proliferation and cytokine production. Similarly, a MUC16 × CD28 BsAb enhances the antitumor efficacy of CD3 BsAbs in a xenograft model of ovarian cancer [30]. In a different study, PSMA × CD28 and EGFR × CD28 BsAbs enhanced the antitumor efficacy of the PD-1 blockade in syngeneic and xenograft tumor models and did not induce systemic T cell activation [29,30]. Other tumor antigen-targeting BsAbs, including B7-H3 × CD28 and PD-L1 × CD28, showed similar costimulatory activity [96]. Similar to T-cell engagers, CD28-targeting BsAbs might induce systemic toxicity because of the constitutive expression of CD28 on T cells. Tumor-targeted ICIs have been investigated, such as EGFR × PD-L1 and CSPG4 × PD-L1, to overcome irAEs induced by the indiscriminate activation of T cells through the PD-1 blockade. Both BsAbs show a target antigen-directed PD-1 blockade followed by T-cell activation [97,98]. The BsAb-based ICI approach may represent the next step toward improving tumor selectivity, efficacy, and safety of ICIs in target antigen-overexpressing malignancies.

### 5.2. TNFR Superfamily

Among the members of TNFRSF, 4-1BB (CD137, *tnfrsf9*) is a well-documented potent costimulatory receptor. 4-1BB is upregulated upon T cell activation, and 4-1BB ligand (4-1BBL, CD137L)-driven clustering of 4-1BB promotes cytotoxic function, enhances survival, and induces the formation of immunological memory. Although 4-1BB is highly upregulated in activated T cells, it is widely expressed on other immune cells such as NK cells, regulatory T cells (Tregs), DCs, and activated monocytes, as well as in endothelial cells, and has been reported to activate NK cells and induce adhesion molecules on vasculature [90,91,99]. Tumor-infiltrating lymphocytes (TILs) highly express 4-1BB compared to lymphocytes in normal tissue or PBMCs [100], and 4-1BB-expressing CD8 TILs include tumor-specific T cells [101,102]. The restricted expression of 4-1BB on TILs and its costimulatory activity make 4-1BB a promising target for cancer immunotherapy. There was an expectation for the potent stimulatory activity of 4-1BB, so two agonistic 4-1BB mAbs, urelumab (BMS-663513), and utomilumab (PF-05082566), were developed. Urelumab is the first human 4-1BB-agonistic mAb in the IgG4 backbone and does not block the interaction of 4-1BB/4-1BBL [103]. Utomilumab is an IgG2-based 4-1BB agonist that blocks 4-1BBL binding [104]. Although both Abs showed potent activity in preclinical studies, their clinical development was unsuccessful due to the fatal hepatotoxicity of urelumab and the low efficacy of utomilumab [90,104].

Tumor-restricted activation of 4-1BB by BsAbs has been intensively studied to maintain optimal therapeutic efficacy and safety. Several companies are developing 4-1BB BsAbs composed of tumor antigen-targeting arm(s) and 4-1BB-agonistic arm(s). The common feature of these BsAbs is the lack of 4-1BB-agonistic activity in the absence of TAAs. One example is PRS-343 (HER2 × 4-1BB), composed of an Fc-silenced HER2 mAb fused with a 4-1BB-agonistic anticalin [38]. This BsAb is in phase I/II clinical development and has shown a clinical benefit and safety in HER2^+^ cancer patients [105]. Other examples are RG7827 (FAP × 4-1BBL) and RG6076 (CD19 × 4-1BBL), which are fusion proteins composed of trimeric 4-1BBL in one arm, and FAP or CD19-targeting Fab in the other arm. RG7827 showed synergy with CEA × CD3 BsAb in a xenograft model [106]. The third example is ALG.APV-527 (5T4 × 4-1BB) comprising two 4-1BB-agonistic scFvs and two 5T4-targeting scFvs with a modified Fc domain that minimizes the interaction with the FcγR. ALG.APV-527 enhanced CD8 T cells and NK cells in the presence of 5T4^+^ cells and inhibited tumor growth in a syngeneic bladder cancer model [107]. A novel 4-1BB BsAb (B7-H3 × 4-1BB) was recently developed to activate 4-1BB signaling only in the context of TAA engagement. This is composed of an Fc-silenced anti-B7-H3 mAb fused with 4-1BB-agonistic scFvs to the C-terminus of the heavy chain. B7-H3 × 4-1BB activates terminally differentiated Tim-3^+^ CD8 T cells in the TME and synergizes with the PD-1 blockade without inducing irAEs [108]. In addition, the novel PD-L1 × 4-1BB and Claudin18.2 × 4-1BB codeveloped by ABL Bio and I-Mab are in phase I clinical trials. These novel 4-1BB BsAbs are anticipated to show strong antitumor efficacy with no liver toxicity due to the localized activation of 4-1BB in tumors.

Glucocorticoid-induced TNF receptor-related protein (GITR, *tnfrsf18*) belongs to the TNFRSF and transmits a costimulatory signal upon binding its ligand, GITRL, which is expressed on APCs. GITR is upregulated upon T cell activation and is constitutively expressed on Tregs [91]. Although in vivo GITR activation enhances tumor immunity, irAEs have been observed [109]. Similar to 4-1BB BsAbs, a FAP scFv-targeted GITRL fusion protein was developed to stimulate T cells exclusively in the presence of FAP-expressing tumor cells and cancer-associated fibroblasts. This molecule costimulates T cells and reduces the suppressive function of Tregs [110].

Another interesting concept is the tumor-targeted activation of CD40. CD40 is a potent stimulator of APCs and myeloid cells, and its activation by CD40L has been shown to activate and license APCs to prime cytotoxic T cells, resulting in enhanced tumor immunity. Agonistic CD40 mAbs have shown efficacy, but systemic dose-limiting toxicity similar to agonists targeting other members of the TNFRSF has been observed in clinical trials [99,111]. ABBV-428 (MSLN × CD40) is an MSLN-targeted CD40 agonist BsAb composed of a homodimer of identical chains containing two scFvs and a modified Fc region that cannot induce ADCC. The mouse version of MSLN × CD40 activated B cells and monocyte-derived DCs and showed MSLN-dependent antitumor efficacy without liver toxicity in syngeneic tumor models. ABBV-428 also showed MSLN-directed CD40 activation in APCs and antitumor efficacy in a xenograft model [112] and is undergoing phase I clinical development.

In summary, the goal of developing most tumor-targeted BsAb-based immunomodulators is to increase the therapeutic window by overcoming the toxicity limitations by activating in the TME. The molecular formats of most tumor-targeted immunomodulators consist of IgG-like BsAbs that have modified Fc regions, allowing persistent activity in the system and preventing Fc function-mediated target cell depletion and FcR-mediated activation of the agonistic target. Unlike CD3, expressed in all T cells, the target for tumor-targeted immunomodulators, especially 4-1BB, is expressed only in the activated T cells. Thus, 4-1BB BsAbs have been developed as the second generation of T-cell engagers.

## 6. Dual Immunomodulators

Among the wide range of immune checkpoint signaling pathways, the PD-1/PD-L1 pathway became the most significant regulator of T cell activity and was extensively investigated [8]. Many dual immune-modulators take an add-on approach of second inhibitory pathways on the PD-1/PD-L1 pathway. The rationale for these BsAbs is their induced expression in the TME and their contribution to primary or acquired resistance to PD-(L)1-targeting therapeutics. Most dual immunomodulators contain the Fc domain to extend their half-life and show enhanced efficacy compared to the combination of mAbs in preclinical studies. Currently, some BsAbs are being evaluated in the clinic (Table 3).

### 6.1. Dual Blockade of Immunosuppressive Targets

The most established dual immunomodulating BsAbs bind simultaneously to PD-(L)1 and other immune inhibitory molecules, such as CTLA-4, LAG-3, Tim-3, TIGIT, TGF-β, and CD73. This type of BsAb is designed to (1) maximize the antitumor activity of T cells by blocking more than one coinhibitory receptor or modifying immunosuppressive TME, and (2) minimize irAEs occasionally observed in the combination of two different ICIs. Although more comprehensive studies are required, dual IC blocking BsAb has shown superior activity in preclinical studies compared to combination therapy. For example, MEDI5752 (PD-1 × CTLA-4) is a monovalent BsAb that inhibits PD-1 and CTLA-4 with reduced Fc function and was designed to block CTLA-4-mediated inhibition on PD-1^+^ T cells. Upon binding to targets, MEDI5752 is rapidly internalized and leads to the subsequent degradation of PD-1, which is not observed with mAbs [113,114]. BsAbs targeting LAG-3, Tim-3, and TIGIT with PD-(L)1 have been developed actively because of better efficacy than combinations in preclinical studies [115].

Another example of the dual blockade is the combination of ICIs with TME modulators, such as TGF-β and CD73. M7824 (PD-L1 × TGF-β, Bintrafusp Alfa) is a bifunctional fusion protein composed of a human IgG1 mAb against PD-L1 fused with two extracellular domains of TGF-βRII, which functions as a TGF-β “trap” for all three TGF-β isoforms. Because the upregulation of TGF-β signaling-associated genes has been linked to anti-PD-1 resistance in metastatic melanoma, M7824 showed better efficacy in preclinical studies [116]. CD73 catalyzes AMP breakdown to adenosine and suppresses immune activation through the A_2A_ receptor [117]. Thus, PD-(L)1 blockade could be combined with the inhibition of the purinergic signaling pathway regulated by a series of nucleotidases, including CD73.

### 6.2. Simultaneous Targeting of Coinhibitory and Costimulatory Pathways

ATOR-1015 (OX40 × CTLA-4) is a tetravalent dual immuno-modulator targeting OX40 and CTLA-4. It consists of a clustering-dependent OX40 agonistic human IgG1 antibody fused with the optimized version of the Ig-like V-type domain of CD86, which is a natural CTLA-4 ligand that binds to CTLA-4 with high affinity while having low affinity for CD28, another counter-receptor. This BsAb may preferentially localize to the tumor because both OX40 and CTLA-4 are highly expressed on activated T cells and Tregs in the TME. ATOR-1015 showed costimulatory activity and enhanced the Fc-mediated ADCC effect on Tregs compared to mAbs [118].

### 6.3. Dual Stimulation of Costimulatory Pathways

Another strategy in immunomodulation is the dual stimulation of costimulatory receptors on T cells. FS120 (OX40 × 4-1BB) is a tetravalent dual agonist, IgG-based BsAb targeting OX40 and 4-1BB through 4-1BB-binding Fab and OX40-binding Fcab. Both OX40 and 4-1BB stimulate T cell proliferation and activation, but OX40 stimulation preferentially activates CD4 T cells, while 4-1BB stimulation preferentially activates CD8 T cells [99]. Several clinical trials have evaluated agonist mAbs to OX40 or 4-1BB; however, both agonists showed limited efficacy or liver toxicity. FS120 activates CD4 and CD8 T cells through dual binding of OX40 and 4-1BB [119]. As the clinical development of tumor-targeted agonistic BsAbs targeting costimulatory pathways progresses, interest in developing such dual costimulators was expanded.

## 7. Dual Tumor-Targeting BsAbs

In addition to immunomodulating BsAbs, dual tumor-targeting by BsAbs may offer simultaneous modulation of two functional pathways in the TME, improved payload delivery, and tumor-restricted tumor cell lysis.

### 7.1. Tumor Receptor Tyrosine Kinase Blockade

Cancer involves multiple disease-driving proteins and pathway crosstalk, which supports a complex molecular network. Identification of these factors has led to the clinical development of targeted therapies for various malignancies. Targeted inhibition of oncogenic receptor tyrosine kinases (RTKs), such as members of the ErbB family, including EGFR and HER2, has been successful in the clinic for over two decades [120]. However, the development of acquired drug resistance is a significant limitation of such therapies. This drug resistance often involves bypassing targeted receptor inhibition by activating crosstalking pathways, such as the heregulin/HER3, hepatocyte growth factor (HGF)/MET, or IGF-1R pathways [121,122]. Several BsAbs targeting two RTKs have been developed to overcome drug resistance, several of which are in clinical trials (Table 4). These BsAbs usually contain the Fc domain for sustained blockade with an extended half-life, and some of them show enhanced ADCC activity through Fc modification.

### 7.2. Angiogenesis Inhibition

Cancer is characterized by some unique properties, such as vascular abnormalities, abnormal tumor immune microenvironment, and hypoxia. The inhibition of angiogenesis to normalize tortuous tumor vasculature, which is essential for tumor development and progression, enabling the alleviation of hypoxia and efficient infiltration of immune cells, has been suggested as a promising approach in cancer therapy [123,124]. Tumor angiogenesis is regulated by multiple angiogenic factors, such as VEGF-A/VEGFR2, Ang-2, angiopoietin-2/angiopoietin receptor, and DLL4/Notch signaling. Therefore, targeting redundancy for multiple angiogenic factors by BsAbs represents an area of interest for cancer therapy (Table 4).

### 7.3. Improved Delivery of Payloads

Antibody-drug conjugates (ADCs) deliver conjugated payloads directly to the tumor by binding to target antigens, internalizing, and exerting their effects [125]. Because there are no apparent tumor-specific antibodies or tumor-selective antigens that do not always internalize well, BsAbs are used for improved payload delivery. This type of BsAbs enhances internalization by targeting TAA and CD63 [126], which is involved in shuttling between the plasma membrane and lysosomal compartment, or non-overlapping epitopes on the same TAA [127]. In addition, ZW49, a HER2-targeted biparatopic ADC for the treatment of HER2-expressing cancers, is in clinical development [128].

### 7.4. Tumor-Targeted Tumor Cell Lysis

The last strategy for dual tumor-targeting BsAbs is tumor-targeted tumor cell elimination. These BsAbs target TAAs and the CD47/SIRPα pathway. CD47 is often overexpressed on tumor cells to evade ADCP by macrophages and other myeloid cells [129]. To block the activity of CD47 in the TME selectively, dual targeting BsAbs with CD47 and CD19/CD20/mesothelin have been developed [130]. In addition, the apoptotic pathway is triggered by the binding of pro-apoptotic proteins, including FasL (CD95L) and TRAIL, to death receptors (DRs) such as Fas receptor (CD95), DR4 (TRAIL receptor 1, TRAIL-R1), and DR5 (TRAIL receptor 2, TRAIL-R2) [131]. Tumor-targeted apoptosis can be activated by DR agonistic BsAbs [132,133]. These BsAbs are expected to overcome the safety or efficacy issues of the respective mAbs, and some BsAbs have been clinically evaluated (Table 4).

## 8. Concluding Remarks

To date, many BsAbs in various forms have been developed and evaluated in clinical trials. Although there are two FDA-approved BsAbs, blinatumomab and catumaxomab, most BsAbs are in the preclinical and early clinical stages. For the success of BsAbs in the clinic, they should overcome the limitations of current immunotherapy and provide better efficacy and safety. Immune effector cell-redirecting BsAbs occupy the most considerable portion of the total BsAbs under development, and among them, T cell-engagers are being developed most actively with more than 60 molecules in clinical development. Safety issues, such as CRS and neurotoxicity, and low efficacy in solid tumors remain critical challenges for T-cell engagers. Some strategies, such as affinity modulation of the CD3-targeting arm, alteration of format, and tumor-specific activation of the CD3-targeting arm, have been suggested for reducing adverse events, but clinical validation is required. NK-cell engagers or 4-1BB-based T-cell engagers may have a better safety profile than CD3-based T-cell engagers, but they are still in the early phase of clinical development and should be validated in human patients.

Tumor-targeted immunomodulators are a class of BsAbs that have been developed to overcome the limitations of their respective immunomodulating mAbs, such as agonistic antibodies. A novel strategy of BsAbs with tumor-specific targeting immunomodulators is designed to activate T cells only in the TME. This type of BsAb is designed to show activity only in the presence of TAAs, which may result in a better safety profile than agonistic mAbs by reducing the risk of irAEs, including liver toxicity. Although both strategies have shown the desired efficacy and safety in early development, it is crucial to select appropriate TAAs to ensure tumor-specific activation and reduce irAEs.

Dual immunomodulators have been developed to exhibit synergistic effects by simultaneously regulating the activities of two different immunomodulatory pathways. These BsAbs are dominated by ICIs, targeting the PD-1/PD-L1 pathway and other immunosuppressive targets, such as LAG-3, Tim-3, TGF-β, and purinergic pathways. There is a general interest in ICIs because of the clinical success of the PD-1/PD-L1 blockade. Thus, it is expected that the development of multiple dual immunostimulators will increase. However, the risk of toxicity due to strong immune activation needs to be monitored.

Finally, dual tumor-targeting BsAbs are designed for the following specific purposes: simultaneous blockade of crosstalking pathways, enhanced drug delivery, and tumor-targeted destruction of tumor cells. To overcome the acquired drug resistance and tumor escape from mAb-based targeted therapy, strategies for the simultaneous inhibition of different signaling pathways using BsAb and simultaneous blockade of two key signaling factors have been proposed. For enhanced efficacy in the TME, several BsAb approaches have been evaluated, typically targeting TAA and effector molecules such as CD63, CD47, pro-apoptotic proteins, and complement-regulatory proteins.

To summarize, the field of immunotherapy using BsAbs is growing rapidly, and the cases of preclinical stage and clinical trials are increasing. Future clinical trials will improve our understanding of the potential and safety profiles of BsAbs. Eventually, BsAbs could provide the next generation of new treatment options for cancer patients.

## Figures and Tables

**Figure 1 vaccines-09-00724-f001:**
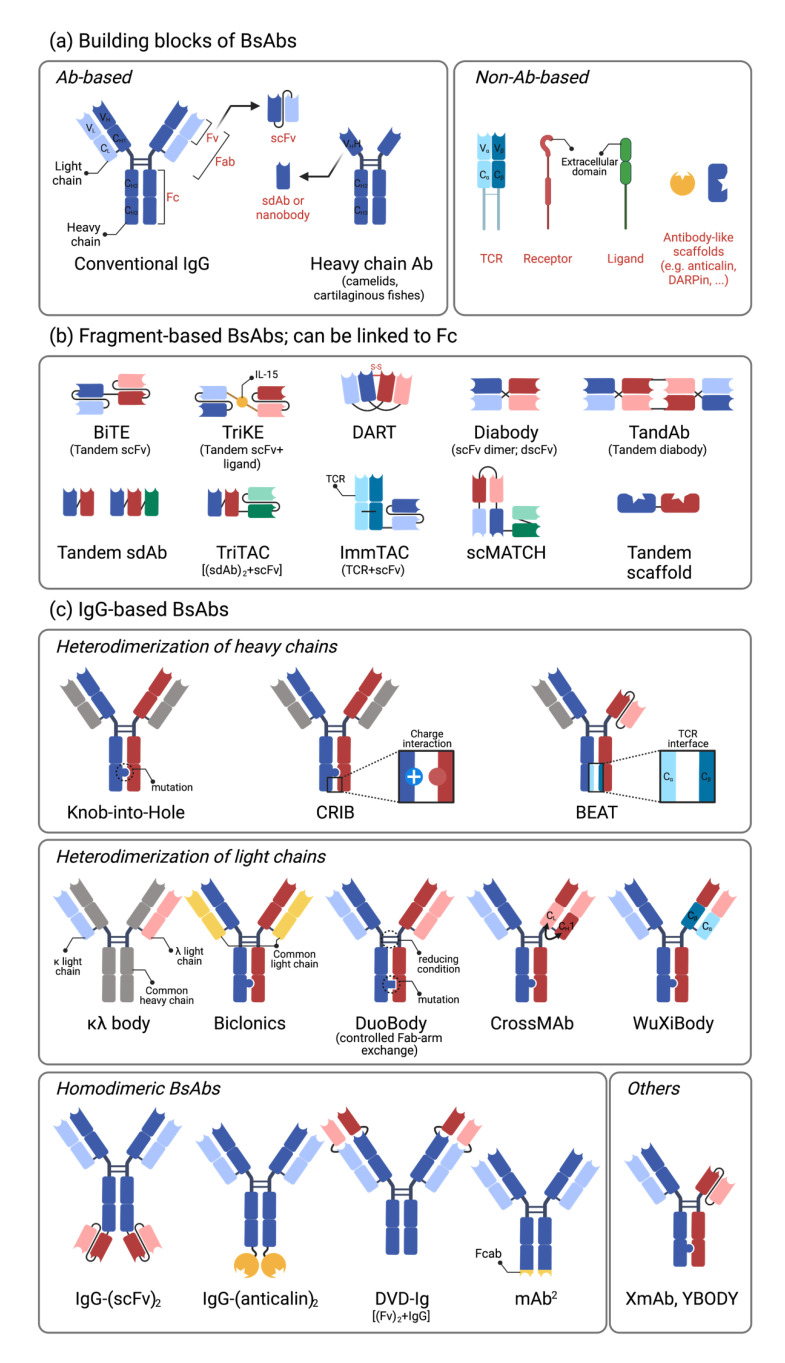
The building blocks and formats of bispecific antibodies. (**a**). The building blocks of BsAbs. Most BsAbs consist of antibody-based fragments (Fv, Fab, Fc, scFv, and sdAb), and some BsAbs include non-antibody-based proteins such as TCR, extracellular domains from receptors or ligands, and antibody-like scaffolds. Fv (V_L_ + V_H_), Fab (V_L_-C_L_ + V_H_-C_H_1), Fc (C_H_2-C_H_3), and scFv (V_L_-V_H_) are derived from the conventional antibody. sdAb (V_H_H) or nanobody is derived from the heavy chain antibody in camelids or cartilaginous fishes. (**b**). The examples of fragment-based BsAbs. These types of BsAbs are developed by linking the building blocks in a. (**c**). The example of IgG-based BsAbs. These BsAbs are developed by pairing two different heavy chains and light chains (heterodimeric BsAb) or linking the building blocks to IgG (homodimeric BsAb). Various strategies are used for proper pairing; knob-into-hole, CRIB, and BEAT for heterodimerization of heavy chains, κλ body, Biclonics, DuoBody, CrossMAb, and WuXiBody for heterodimerization of light chains. Abbreviations: BiTE; bispecific T-cell engager; TriKE, trispecific killer cell engager; DART, dual-affinity retargeting; TandAb, tandem diabody; TriTAC, trispecific T cell-activating construct; ImmTAC, immune mobilizing monoclonal TCR against cancer; scMATCH, single-chain multispecific antibody-based therapeutics by cognate heterodimerization; CRIB, charge repulsion induced bispecific; BEAT, bispecific engagement by antibodies based on the TCR; κλ body, kappa lambda body; DVD-Ig, dual-variable-domain immunoglobulin; Fcab, Fc region with an antigen-binding site.

**Figure 2 vaccines-09-00724-f002:**
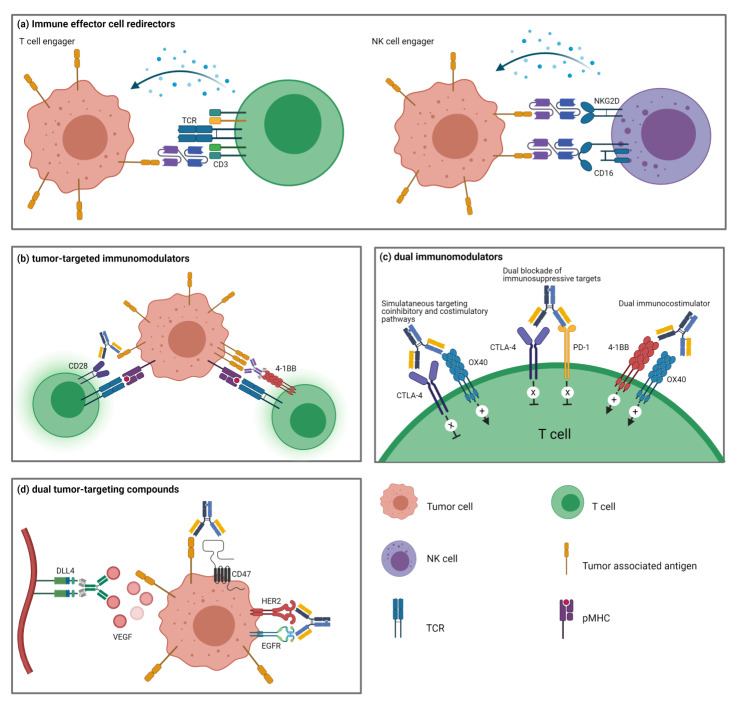
The types of bispecific therapeutics in immuno-oncology. Bispecific therapeutics are grouped into four major categories based on the biological targets and mode of actions. (**a**). Immune effector cell redirectors engage tumor cell and immune effector cells by binding to tumor-associated antigen and immune cell receptors such as CD3, CD16, or NKG2D. (**b**). Tumor-targeted immunomodulators simultaneously bind to cosignaling molecules (CD28 and 4-1BB) on T cells and tumor cells, inducing tumor-specific activation of T cells. (**c**). Dual immunomodulators regulate the activity of two different immunomodulatory pathways by dual targeting of cosignaling pathways. (**d**). Dual tumor-targeting BsAbs modulate two functional pathways in the TME by inhibiting angiogenesis, RTK activity, or CD47 binding.

**Table 1 vaccines-09-00724-t001:** Immune modulating bispecific antibody formats in clinical trials for cancer therapy.

Category	Antibody Format/Platform	Structure	Company	Product	Target
Single domain antibody (sdAb)-based BsAbs [16,17,18]		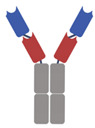	Alphamab	KN046	PD-L1 × CTLA-4
Inhibrix	INBRX-105	PD-L1 × 4-1BB
Single-chain variable fragment (scFv)-based BsAbs [19,20,21,22,23,24,25,26,27]	Bispecific T-cell Engager (BiTE)	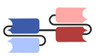	Amgen	Blinatumomab (Blincyto)	CD19 × CD3
AMG330	CD33 × CD3
AMG420	BCMA × CD3
Bayor	BAY2010112	PSMA × CD3
BiTE-Fc/Half-lifeextended BiTE(HLE BiTE)	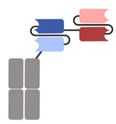	Amgen	AMG701 (Pavurutamab)	BCMA × CD3
AMG673	CD33 × CD3
AMG757	DLL3 × CD3
Dual affinity retargeting (DART)	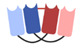	Macrogenics	MGD006 (Flotetuzumab)	CD123 × CD3
DART-Fc	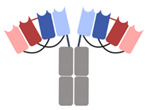	Macrogenics	MGD013 (Tebotelimab)	PD-1 × LAG-3
MGD019	PD-1 × CTLA-4
Tandem Diabody (TandAb)	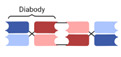	Affimed	AFM13	CD30 × CD16A
Amphivena	AMV564	CD33 × CD3
Trispecific KillerEngager (TriKE)	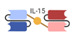	GT Biopharma	GTB-3550	CD33 × CD16, IL-15
IgG-based,heterodimeric bispecifics [28,29,30,31,32,33,34]	Common light chain(Biclonics,Veloci-Bi)	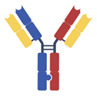	Merus	MCLA-145	PD-L1 × 4-1BB
Regeneron	REGN1979	CD20 × CD3
REGN5678	PSMA × CD28
DuoBody; Controlled Fam-arm exchange	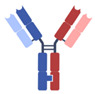	Janssen	Teclistamab(JNJ-64007957)	BCMA × CD3
Genmab-Abbvie	Epcoritamab(GEN3013)	CD20 × CD3
Genmab-Biontech	BNT312(GEN1042)	CD40 × 4-1BB
BNT311(GEN1046)	PD-L1 × 4-1BB
1 + 1 CrossMAb	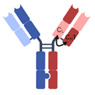	Roche	RO7247669	PD-1 × LAG-3
Mosunetuzumab	CD20 × CD3
Cevostamab	FcRH5 × CD3
2 + 1 CrossMAb	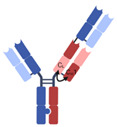	Roche	Glofitamab	CD20 × CD3
RG7802(RO6958688)	CEA × CD3
Celgene	CC-93269	BCMA × CD3
IgG-based,homodimeric bispecifics [35,36,37,38,39]	Grabody, IgG-scFv	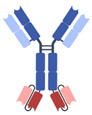	ABL Bio	ABL503	PD-L1 × 4-1BB
ABL111	Claudin 18.2 × 4-1BB
IgG-anticalin fusion protein	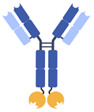	Pieris	PRS-343	HER2 × 4-1BB
Fc region with antigen binding (Fcab), mAb^2^	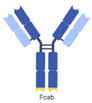	F-Star	FS118	PD-L1 × LAG-3
Others [40,41,42,43]	XmAb, Fab + scFv + Fc	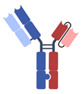	Xencor	Vibecotamab (XmAb 14045)	CD123 × CD3
XmAb20717	PD-1 × CTLA-4
Bispecific Engagement by Antibodies based on TCR (BEAT)	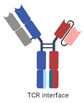	Glenmark Pharmaceuticals	GBR1342	CD38 × CD3

**Table 2 vaccines-09-00724-t002:** Immune cell-engagers in clinical stages.

Name	Targets	Developer	Format	Highest Phase	Clinical Trials
AFM13	CD30 × CD16A	Affimed GmbH	Fv + (Fv)_2_ + Fv, 2 + 2 TandAb	II	NCT04074746 and 5 studies
AFM24	EGFR × CD16A	Affimed GmbH	Fv + (Fv)_2_ + Fv, 2 + 2 TandAb	I/II	NCT04259450
AFM26, RO7297089	BCMA × CD16A	Affimed GmbH, Genentech	Fv + (Fv)_2_ + Fv, 2 + 2 TandAb	I	NCT04434469
Solitomab, AMG110, MT110	EpCAM × CD3	Amgen	scFv + scFv, 1 + 1 BiTE	I	NCT00635596
AMG160	PSMA × CD3	Amgen	scFv + scFv + Fc, 1 + 1 HLE BiTE	I	NCT03792841 NCT04631601
AMG199	MUC17 × CD3	Amgen	scFv + scFv + Fc, 1 + 1 HLE BiTE	I	NCT04117958
AMG330	CD33 × CD3	Amgen	scFv + scFv, 1 + 1 BiTE	I	NCT02520427 NCT04478695
AMG427	FLT3 × CD3	Amgen	scFv + scFv + Fc, 1 + 1 HLE BiTE	I	NCT03541369
AMG562	CD19 × CD3	Amgen	scFv + scFv, 1 + 1 BiTE	I	NCT03571828
AMG596	EGFRvIII × CD3	Amgen	scFv + scFv, 1 + 1 BiTE	I	NCT03296696
AMG673	CD33 × CD3	Amgen	scFv + scFv + Fc, 1+ 1 HLE BiTE	I	NCT03224819
Pavurutamab, AMG701	BCMA × CD3	Amgen	scFv + scFv + Fc, 1 + 1 HLE BiTE	I	NCT03287908
AMG757	DLL3 × CD3	Amgen	scFv + scFv + Fc, 1 + 1 HLE BiTE	I	NCT03541369
AMG910	CLDN18.2 × CD3	Amgen	scFv + scFv + Fc, 1 + 1 HLE BiTE	I	NCT04260191
Pasotuxizumab, AMG212, BAY2010112	PSMA × CD3	Amgen, Bayer AG	scFv + scFv, 1 + 1 BiTE	I	NCT01723475
AMG420, BI836909	BCMA × CD3	Amgen, Boehringer Ingelheim	scFv + scFv, 1 + 1 BiTE	I	NCT03836053
AMG424	CD38 × CD3	Amgen, Xencor	Fab + scFv + Fc, 1 + 1 XmAb	I	NCT03445663
AMG509	STEAP1 × CD3	Amgen, Xencor	(Fab)_2_ + scFv + Fc, 2 + 1 XmAb	I	NCT04221542
AMV564	CD33 × CD3	Amphivena Therapeutics	Fv + (Fv)_2_ + Fv, 2 + 2 TandAb	I	NCT04128423 and 2 studies
APVO436	CD123 × CD3	Aptevo Therapeutics	(scFv)_2_ + (scFv)_2_ + Fc, 2 + 2 ADAPTIR	I	NCT03647800
CC-93269, EM801	BCMA × CD3	Celgene	Fab + (Fab)_2_ + Fc, 2 + 1 CrossMAb	I	NCT03486067
ERY974	Glypican-3 × CD3	Chugai Pharmaceutical	Fab + Fab + Fc, 1 + 1 ART-Ig	I	NCT02748837
A-319	CD19 × CD3	Evive Biotech	scFv + Fab, 1 + 1	I	NCT04056975
GEM333	CD33 × CD3	GEMoaB Monoclonals	scFv + scFv, 1 + 1	I	NCT03516760
GEM3PSCA	PSCA × CD3	GEMoaB Monoclonals	scFv + scFv, 1 + 1	I	NCT03927573
RG6160, RO7187797, BFCR4350A	FcRH5 × CD3	Genentech	Fab + Fab + Fc, 1 + 1	I	NCT03275103
RG6194, BTRC4017A	HER2 × CD3	Genentech	undisclosed	I	NCT03448042
RG6296, RO7297089	BCMA × CD16A	Genentech	Fab + Fab + Fc, 1 + 1	I	NCT04434469
Mosunetuzumab, RG7828, RO7030816, BTCT4465A	CD20 × CD3	Genentech, Roche, Chugai	Fab + Fab + Fc, 1 + 1	II	NCT03677154 and 8 studies
GEN1044	5T4 × CD3	Genmab, Abbvie	Fab + Fab + Fc, 1 + 1 DuoBody	I	NCT04424641
Epcoritamab, GEN3013	CD20 × CD3	Genmab, Abbvie	Fab + Fab + Fc, 1 + 1 DuoBody	III	NCT03625037 and 5 studies
GTB-3550, OXS-3550	CD33 × CD16, IL-15	GT Biopharma	scFv + ligand + scFv, 1 + 1 + 1 TriKE	I/II	NCT03214666
HPN424	PSMA × HSA × CD3	Harpoon Therapeutics	sdAb + sdAb + scFv, 1 + 1 + 1 TriTAC	I/II	NCT03577028
ISB1302, GBR1302	HER2 × CD3	Ichnos Sciences, Glenmark Pharmaceuticals	Fab + scFv + Fc, 1 + 1 BEAT	I/II	NCT02829372 NCT03983395
ISB1342, GBR1342	CD38 × CD3	Ichnos Sciences, Glenmark Pharmaceuticals	Fab + scFv + Fc, 1 + 1 BEAT	I/II	NCT03309111
IGM-2323	CD20 × CD3	IGM Biosciences	IgM + scFv, 10 + 1	I	NCT04082936
Tebentafusp, IMCgp100	gp100/HLA-A * 02:01 × CD3	Immunocore	TCR + scFv, 1 + 1 ImmTAC	III	NCT03070392 and 5 studies
IMC-F106C	PRAME/HLA-A * 02:01 × CD3	Immunocore	TCR + scFv, 1 + 1 ImmTAC	I/II	NCT04262466
IMC-C103C	MAGE-A4/HLA-A * 02:01 × CD3	Immunocore, Genentech	TCR + scFv, 1 + 1 ImmTAC	I/II	NCT03973333
IMCnyeso, GSK01	NY-ESO-1/HLA-A * 02:01 × CD3	Immunocore, GlaxoSmithKline	TCR + scFv, 1 + 1 ImmTAC	I/II	NCT03515551
JNJ-63709178	CD123 × CD3	Janssen Research & Development	Fab + Fab + Fc, 1 + 1 DuoBody	I	NCT02715011
JNJ-63898081	PSMA × CD3	Janssen Research & Development	Fab + Fab + Fc, 1 + 1 DuoBody	I	NCT03926013
Teclistamab, JNJ-64007957	BCMA × CD3	Janssen Research & Development	Fab + Fab + Fc, 1 + 1 DuoBody	II	NCT04557098 and 5 studies
Talquetamab, JNJ-64407564	GPRC5D × CD3	Janssen Research & Development	Fab + Fab + Fc, 1 + 1 DuoBody	II	NCT04634552 and 3 studies
JNJ-67571244	CD33 × CD3	Janssen Research & Development	Fab + Fab + Fc, 1 + 1 DuoBody	I	NCT03915379
MGD007	gpA33 × CD3	MacroGenics	Fv + Fv + Fc, 1 + 1 DART-Fc	I	NCT02248805 NCT03531632
Orlotamab, MGD009	B7-H3 × CD3	MacroGenics	Fv + Fv + Fc, 1 + 1 DART-Fc	I	NCT02628535 NCT03406949
Duvortuxizumab, MGD011, JNJ-64052781	CD19 × CD3	MacroGenics, Janssen Research & Development	Fv + Fv, 1 + 1 DART	I	NCT02743546 NCT02454270
Flotetuzumab, MGD006, S80880	CD123 × CD3	MacroGenics, Servier	Fv + Fv, 1 + 1 DART	I	NCT04582864 and 5 studies
Tepoditamab, MCLA-117	CLEC12A × CD3	Merus	Fab + Fab + Fc, 1 + 1 Biclonics	I	NCT03038230
PF-06671008	P-cadherin × CD3	Pfizer	Fv + Fv + Fc, 1 + 1 DART-Fc	I	NCT02659631
PF-06863135	BCMA × CD3	Pfizer	Fab + Fab + Fc, 1 + 1	II	NCT04649359 and 2 studies
Odronextamab, REGN1979	CD20 × CD3	Regeneron	Fab + Fab + Fc, 1 + 1	II	NCT03888105 and 2 studies
REGN5458	BCMA × CD3	Regeneron	Fab + Fab + Fc, 1 + 1	I/II	NCT03761108
REGN5459	BCMA × CD3	Regeneron	Fab + Fab + Fc, 1 + 1	I	NCT04083534
REGN4018	MUC16 × CD3	Regeneron, Sanofi	Fab + Fab + Fc, 1 + 1	I/II	NCT03564340 NCT04590326
Glofitamab, RO7082859, RG6026	CD20 × CD3	Roche	Fab + (Fab)_2_ + Fc, 2 + 1 CrossMAb	III	NCT03075696 and 8 studies
Cibisatamab, RO6958688, RG7802	CEA × CD3	Roche, Genentech	Fab + (Fab)_2_ + Fc, 2 + 1 CrossMAb	I	NCT02650713 and 3 studies
SAR440234	CD123 × CD3	Sanofi	Fab + Fv + Fc, 1 + 1	I/II	NCT03594955
TNB-383B	BCMA × CD3	TeneoBio, AbbVie	sdAb + Fab + Fc, 1 + 1	I	NCT03933735
M802	HER2 × CD3	Wuhan YZY Biopharma	Fab + scFv + Fc, 1 + 1 YBODY	I	NCT04501770
Plamotamab, Xmab13676	CD20 × CD3	Xencor	Fab + scFv + Fc, 1 + 1 XmAb	I	NCT02924402
Tidutamab, Xmab18087	SSTR2 × CD3	Xencor	Fab + scFv + Fc, 1 + 1 XmAb	I/II	NCT03411915 NCT04590781
Vibecotamab, Xmab14045	CD123 × CD3	Xencor, Novartis	Fab + scFv + Fc, 1 + 1 XmAb	I	NCT02730312
Nivatrotamab	GD2 × CD3	Y-mAbs	IgG + (scFv)_2,_ 2 + 2 BiClone	I/II	NCT04750239

**Table 3 vaccines-09-00724-t003:** Tumor-targeted or Dual immunomodulators in clinical stages.

Name	Targets	Developer	Format	Highest Phase	Clinical Trials
ABBV-428	MSLN × CD40	AbbVie	(scFv)_2_ + (scFv)_2_ + Fc, 2 + 2	I	NCT02955251
ABL503	PD-L1 × 4-1BB	ABL Bio, I-MAB	IgG + (scFv)_2_, 2 + 2	I	NCT04762641
AK104	PD-1 × CTLA-4	Akeso Biopharma	IgG + (scFv)_2_, 2 + 2	II	NCT04172454 and 13 studies
ATOR-1015, ADC-1015	OX40 × CTLA-4	Alligator Bioscience	IgG + (ligand)_2_, 2 + 2	I	NCT03782467
AMG506, MP0310	FAP × 4-1BB × HSA	Molecular Partners AG, Amgen	(DARPin)_3_, 1 + 1 + 1	I	NCT04049903
CDX-527	PD-1 × CD40	Celldex	IgG + (scFv)_2_, 2 + 2	I	NCT04440943
LY3415244	PD-L1 × Tim-3	Eli Lilly	Fab + Fab + Fc, 1 + 1	I	NCT03752177
LY3434172	PD-1 × PD-L1	Eli Lilly	Fab + Fab + Fc, 1 + 1	I	NCT03936959
FS118	PD-L1 × LAG-3	F-star	IgG with Fcab, 2 + 2 mAb^2^	I/II	NCT03440437
FS120	OX40 × 4-1BB	F-star	IgG with Fcab, 2 + 2 mAb^2^	I	NCT04648202
FS222	PD-L1 × 4-1BB	F-star	IgG with Fcab, 2 + 2 mAb^2^	I	NCT04740424
GEN1042	CD40 × 4-1BB	Genmab, BioNTech	Fab + Fab + Fc, 1 + 1 DuoBody	I	NCT04083599
GEN1046, BNT311	PD-L1 × 4-1BB	Genmab, BioNTech	Fab + Fab + Fc, 1 + 1 DuoBody	I/II	NCT03917381
GS-1423, AGEN1423	CD73 × TGF-b	Gilead Sciences, Agenus	IgG + TGFβ receptor, 2 + 2	I	NCT03954704
INBRX-105, ES101	PD-L1 × 4-1BB	Inhibrx, Elpiscience Biopharma	(sdAb)_2_ + (sdAb)_2_ + Fc, 2 + 2	I	NCT03809624 NCT04009460
IBI318	PD-1 × PD-L1	Innovent Biologics	Fab + Fab + Fc, 1 + 1	II	NCT04777084 and 5 studies
KN046	PD-L1 × CTLA-4	Jiangsu Alphamab Biopharmaceuticals	(sdAb)_2_ + (sdAb)_2_ + Fc, 2 + 2	III	NCT04040699 and 15 studies
SHR-1701	PD-L1 × TGF-b	Jiangsu HengRui Medicine	IgG + TGFβ receptor, 2 + 2	II	NCT04650633 and 11 studies
MGD019	PD-1 × CTLA-4	Macrogenics	(Fv)_2_ + (Fv)_2_ + Fc, 2 + 2 DART-Fc	I	NCT03761017
Tebotelimab, MGD013	PD-1 × LAG-3	Macrogenics, Zai Lab	(Fv)_2_ + (Fv)_2_ + Fc, 2 + 2 DART-Fc	II/III	NCT04212221 and 6 studies
MEDI5752	PD-1 × CTLA-4	MedImmune	Fab + Fab + Fc, 1 + 1	I	NCT04522323 and 2 studies
Bintrafusp Alfa, M7824	PD-L1 × TGF-b	Merck KGaA	IgG + TGFβ receptor, 2 + 2	III	NCT03631706 and 38 studies
MCLA-145	PD-L1 × 4-1BB	Merus, Incyte	Fab + Fab + Fc, 1 + 1 Biclonics	I	NCT03922204
NM21-1480	PD-L1 × 4-1BB × HSA	Numab	Fv + Fv + Fv, 1 + 1 + 1 scMATCH3	I/II	NCT04442126
PRS-343	HER2 × 4-1BB	Pieris Pharmaceuticals	IgG + (anticalin)_2_, 2 + 2	I	NCT03330561 NCT03650348
REGN5678	PSMA × CD28	Regeneron	Fab + Fab + Fc, 1 + 1	I/II	NCT03972657
RG6139, RO7247669	PD-1 × LAG-3	Roche	Fab + Fab + Fc, 1 + 1 CrossMAb	II	NCT04785820 NCT04140500
RG7769, RO7121661	PD-1 × Tim-3	Roche	Fab + Fab + Fc, 1 + 1 CrossMAb	II	NCT04785820 and 2 studies
Xmab20717	PD-1 × CTLA-4	Xencor	scFv + Fab + Fc, 1 + 1 Xtend XmAb	I	NCT03517488
Xmab22841	CTLA-4 × LAG-3	Xencor	scFv + Fab + Fc, 1 + 1 Xtend XmAb	I	NCT03849469
Xmab23104	PD-1 × ICOS	Xencor	scFv + Fab + Fc, 1 + 1 Xtend XmAb	I	NCT03752398

**Table 4 vaccines-09-00724-t004:** Dual tumor-targeting BsAbs in clinical stages.

Name	Targets	Developer	Format	Highest Phase	Clinical Trials
Dilpacimab, ABT-165	DLL4 × VEGF	AbbVie	(Fv)_2_ + IgG, 2 + 2 DVD-Ig	II	NCT01946074 NCT03368859
ABL001, NOV1501, TR009	DLL4 × VEGF	ABL Bio	IgG + (scFv)_2_, 2 + 2	I/II	NCT03292783 NCT04492033
BI836880	VEGF × Ang-2	Boehringer Ingelheim	sdAb + sdAb + albumin, 1 + 1	II	NCT03861234 and 5 studies
BI905677	LRP5/6	Boehringer Ingelheim	sdAb + sdAb + albumin, 1 + 1	I	NCT03604445
AK112	VEGF × PD-1	Akesobio	IgG + (scFv)_2_, 2 + 2	I/II	NCT04736823 and 2 studies
KN026	HER2 × HER2 (Biparatopic)	Jiangsu Alphamab Pharmaceuticals	Fab + Fab + Fc, 1 + 1	II	NCT04521179 and 6 studies
MBS301	HER2 × HER2 (Biparatopic)	Beijing Mabworks Biotech	Fab + Fab + Fc, 1 + 1 Fab-arm exchange	I	NCT03842085
LY3164530	MET × EGFR	Eli Lilly and Co	Fab + Fab + Fc, 1 + 1 orthoFab-IgG	I	NCT02221882
EMB-01	cMET × EGFR	Epimab Biotherapeutics	(Fab)_2_ + IgG, 2 + 2 FIT-Ig	I/II	NCT03797391
Duligotuzumab, MEHD7945A, RO5541078, RG7597	EGFR × HER3	Genentech	Fab + Fab + Fc, 1 + 1	II	NCT01986166 and 4 studies
Amivantamab, JNJ-61186372	cMET × EGFR	Genmab, Janssen	Fab + Fab + Fc, 1 + 1 DuoBody	III	NCT04077463 and 5 studies
HX009	PD-1 × CD47	HanxBio	IgG + (ligand)_2_, 2 + 2	I	NCT04097769
IMM0306	CD20 × CD47	ImmuneOnco	IgG + (ligand)_2_, 2 + 2	I	CTR20192612
IBI322	PD-L1 × CD47	Innovent	sdAb + Fab + Fc, 1 + 1	I	NCT04795128 and 2 studies
MM-111	HER2 × HER3	Merrimack Pharmaceuticals	(scFv)_2_ + albumin, 1 + 1	II	NCT01097460 and 3 studies
MM-141	IGF-1R × HER3	Merrimack Pharmaceuticals	IgG + (scFv)_2_, 2 + 2	II	NCT02538627 and 2 studies
Zenocutuzumab, MCLA-128,PB4188	HER2 × HER3	Merus	Fab + Fab + Fc, 1 + 1 Biclonics	II	NCT02912949 and 2 studies
MP0250	HGF × VEGF × HSA	Molecular Partners AG	(DARPin)_4_, 1 + 1 + 2	I/II	NCT03136653 and 2 studies
MP0274	HER2 × HER2 (Biparatopic)	Molecular Partners AG	(DARPin)_4_, 1 + 1 + 2	I	NCT03084926
TG-1801, NI-1701	CD19 × CD47	NovImmune, TG Therapeutics	Fab + Fab + Fc, 1 + 1 κλ body	I	NCT03804996 NCT04806035
Navicixizumab, OMP-305B83	DLL4 × VEGF	OncoMed	Fab + Fab + Fc, 1 + 1 common LC	I	NCT03035253 and 2 studies
REGN5093	MET × MET (Biparatopic)	Regeneron	Fab + Fab + Fc, 1 + 1	I/II	NCT04077099
Vanucizumab, RG7221, RO5520985	Ang-2 × VEGF-A	Roche	Fab + Fab + Fc, 1 + 1 CrossMAb	II	NCT02665416 and 4 studies
Zanidatamab, ZW25	HER2 × HER2 (Biparatopic)	Zymeworks	Fab + scFv + Fc, 1 + 1	II	NCT04224272 and 6 studies

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
