# Peer review of "Bispecific Antibodies: A Smart Arsenal for Cancer Immunotherapies"

_vaccines, 2021, doi:10.3390/vaccines9070724_

Round 1

Reviewer 1 Report

This manuscript summarized four classes of BsAbs and their development and applications. BsAbs have the potential to become the next generation of new treatment options for cancer patients. However, it seems that similar works have been present in previous papers. In addition, this manuscript does not show adequate figures and schematics to present the work, and help readers better understand the principles and applications of BsAbs.

    Author Response

    Comments: This manuscript summarized four classes of BsAbs and their development and applications. BsAbs have the potential to become the next generation of new treatment options for cancer patients. However, it seems that similar works have been present in previous papers. In addition, this manuscript does not show adequate figures and schematics to present the work, and help readers better understand the principles and applications of BsAbs.

    Response 1: As a review paper on BsAb, we have tried to summarize recent updates in this field. Therefore, the reviewer may have thought similar to other articles. However, we believe we have been attempting to sort out the recent advancement of BsAb in terms of Ab structure, development, clinical applications, and limitations. I hope the reviewer agrees on this point. According to the reviewer’s comment, we have added a figure summarizing the technology for developing BsAb that is currently used in clinical practice.

    Reviewer 2 Report

    The submitted manuscript reviewed the current progress in bispecific antibodies for cancer immunotherapies. It reviewed the structures, characteristics, types, and clinical trials of bispecific antibodies. 

    Overall, it is detailed and excellent in quality. There are several minor issues to correct before publication.

    • There are red words in table 1.
    • Additional horizontal lines are needed in tables. It is troublesome to separate different rows. e.g.
      • The last column between IGM-2323 and Tebentafusp in table 2
      • The last column between Vanucizumab and Zanidatamab in table 4
    • Table 3 needs reformat.

    Author Response

    The submitted manuscript reviewed the current progress in bispecific antibodies for cancer immunotherapies. It reviewed the structures, characteristics, types, and clinical trials of bispecific antibodies.

    Overall, it is detailed and excellent in quality. There are several minor issues to correct before publication.

    Point 1: There are red words in table 1.

    Response: As the reviewer pointed out, we have changed red words to black.

    Point 2: Additional horizontal lines are needed in tables. It is troublesome to separate different rows. e.g.

    The last column between IGM-2323 and Tebentafusp in table 2

    The last column between Vanucizumab and Zanidatamab in table 4

    Response: As the reviewer pointed out, unlike the first submitted tables, the contents of the tables were center alignment, so it was difficult to distinguish each row. We have modified the contents of the tables to align in the upper left corner.

    Point 3: Table 3 needs reformat.

    Response: As the reviewer pointed out, we have reformatted Table 3.

    Reviewer 3 Report

    The manuscript of You et al. reviews the current knowledge about the use of bispecific antibodies in cancer immunotherapy. The authors describe 4 classes of BsAbs and present an extensive list of examples of these antibodies under development. They also describe the current limitations encounter in the clinical use of BsAbs and strategies to overcome these limitations.

    Major comments:

    The 4 tables are extensive. Please try to select a criterion to resume the tables and/or reduce the number of tables. Example 1: If any of the information of one of the tables was already reviewed in a recent paper, please cite the paper and remove the table from the manuscript and describe only the antibodies not previously reviewed. Example 2: Resume tables, describing only the ones that reached clinical trials II or III.

    Minor comments:

    • Introduction, line 24-36: No reference is cited on this text. Please cite appropriate references.
    • Introduction, line 44-46: Please rephrase this sentence.
    • Introduction, line 54-55: Please add the name of the 2 FDA-approved molecules.
    • Introduction, line 63-64: Please rephrase this sentence.
    • Table 1: No reference is described on this table. Please add if available.
    • Review abbreviations in the tables and on the text, the description of several of them are missing.
    • Page 5, Line 83: Please confirm if the reference 9 is appropriate for this sentence. This reference is from a review manuscript.

    Author Response

    The manuscript of You et al. reviews the current knowledge about the use of bispecific antibodies in cancer immunotherapy. The authors describe 4 classes of BsAbs and present an extensive list of examples of these antibodies under development. They also describe the current limitations encounter in the clinical use of BsAbs and strategies to overcome these limitations.

    Major comments: The 4 tables are extensive. Please try to select a criterion to resume the tables and/or reduce the number of tables. Example 1: If any of the information of one of the tables was already reviewed in a recent paper, please cite the paper and remove the table from the manuscript and describe only the antibodies not previously reviewed. Example 2: Resume tables, describing only the ones that reached clinical trials II or III.

    Response: As the reviewer suggested, we have modified the Tables (reducing the number of NCT identifiers for single BsAb). Although there is some overlap as the reviewer said, we think it is also meaningful to show in this paper the BsAbs from the early stages of development to the most recent ones in Tables. We hope the reviewer understands this point.

    Minor comment 1: Introduction, line 24-36: No reference is cited on this text. Please cite appropriate references.

    Response: As the reviewer pointed out, we have cited references in the introduction.

    Minor comment 2: Introduction, line 44-46: Please rephrase this sentence.

    Response: As the reviewer suggested, we have rephrased the sentence to clarify its meaning.

    Minor comment 3: Introduction, line 54-55: Please add the name of the 2 FDA-approved molecules.

    Response: As the reviewer suggested, we have included the name of the 2 FDA-approved molecules, Blinatumomab and Catumaxomab.

    Minor comment 4: Introduction, line 63-64: Please rephrase this sentence.

    Response: As the reviewer suggested, we have rephrased the sentence to clarify its meaning.

    Minor comment 5: Table 1: No reference is described on this table. Please add if available.

    Response: As the reviewer pointed out, we have included references in Table 1.

    Minor comment 6: Review abbreviations in the tables and on the text, the description of several of them are missing.

    Response: As the reviewer pointed out, we have reviewed the use of abbreviations in the whole manuscript and included appropriate descriptions.

    Minor comment 7: Page 5, Line 83: Please confirm if the reference 9 is appropriate for this sentence. This reference is from a review manuscript.

    Response: As the reviewer pointed out, we have checked the citation of references and changed cited references on that sentence.

    Round 2

    Reviewer 1 Report

    The minor modifications have not significantly improved this manuscript,  therefore do not change my evaluation for this manuscript. 

    Reviewer 3 Report

    The authors aswered to all my comments.

    However, in my opinion the tables could be further improved to be not so extensive.

    In general, the manuscript in the present format is aceptable for publication.